# Tumor Microenvironment, Clinical Features, and Advances in Therapy for Bone Metastasis in Gastric Cancer

**DOI:** 10.3390/cancers14194888

**Published:** 2022-10-06

**Authors:** Pengcheng Sun, Samuel O. Antwi, Kurt Sartorius, Xiao Zheng, Xiaodong Li

**Affiliations:** 1Department of Tumor Biological Treatment, The Third Affiliated Hospital of Soochow University, Changzhou 213004, China; 2Department of Oncology, The Third Affiliated Hospital of Soochow University, Changzhou 213004, China; 3Department of Quantitative Health Sciences, Mayo Clinic, Jacksonville, FL 32224, USA; 4The Africa Hepatopancreatobiliary Cancer Consortium (AHPBCC), Mayo Clinic, Jacksonville, FL 32224, USA; 5School of Laboratory Medicine and Molecular Sciences, College of Health Sciences, University of Kwazulu-Natal, Durban 4041, South Africa; 6UKZN Gastrointestinal Cancer Research Unit, University of Kwazulu-Natal, Durban 4041, South Africa

**Keywords:** gastric cancer, bone metastasis, bone microenvironment, clinical features, skeletal-related events, treatment regimens

## Abstract

**Simple Summary:**

Bone metastasis in gastric cancer (GC-BM) has received little attention because of its rarity, but the increased pain and shortened survival of patients due to the bone destruction and skeletal-related events (SREs) caused by tumors cannot be ignored. Immune cells and molecules such as RANK-L and GFs in complicated bone microenvironments create suitable conditions for tumor cells to proliferate and break the balance of bone circulation, which leads to osteolysis and osteogenesis. Bone radionuclide scans now are the most sensitive examination for diagnosing bone tumors, and bone-related alkaline phosphatase (ALP) also shows a certain significance in the diagnosis of bone tumors. Combined application of chemotherapeutic drugs, targeted drugs, and immune drugs are the main treatments of GC-BM. This review paper intends to discuss the clinical status of GC-BM, as well as its possible molecular pathogenesis pathways and therapeutic options.

**Abstract:**

Gastric cancer (GC) is one of the most malignant neoplasms worldwide, accounting for about 770,000 deaths in 2020. The incidence of gastric cancer bone metastasis (GC-BM) is low, about 0.9–13.4%, and GC patients develop GC-BM because of a suitable bone microenvironment. Osteoblasts, osteoclasts, and tumor cells interact with each other, secreting cytokines such as PTHrP, RANK-L, IL-6, and other growth factors that disrupt the normal bone balance and promote tumor growth. The functions and numbers of immune cells in the bone microenvironment are continuously inhibited, resulting in bone balance disorder due to the cytokines released from destroyed bone and growing tumor cells. Patients with GC-BM are generally younger than 65 years old and they often present with a later stage of the disease, as well as more aggressive tumors. They usually have shorter overall survival (OS) because of the occurrence of skeletal-related events (SREs) and undetected bone destruction due to the untimely bone inspection. Current treatments of GC-BM focus mainly on gastric cancer and SRE-related treatment. This article reviews the clinical features, possible molecular pathogeneses, and the most commonly used diagnostic methods and treatments of bone metastasis in gastric cancer.

## 1. Introduction

As the fifth most common malignant tumor around the world, gastric cancer (GC) has a lower incidence rate and mortality compared with lung cancer and colorectal cancer, but according to estimates from the Global Cancer Observatory (formerly GLOBOCAN), gastric cancer still caused nearly 1.08 million new cases and 770,000 deaths worldwide in 2020 [1]. Although the progress of new treatment methods such as targeted therapy and immunotherapy has diversified the treatment of gastric cancer, the most traditional and effective treatment for gastric cancer is still surgical resection [2].

However, radical surgery is often unable to completely prevent GC recurrence and metastasis to the lymph nodes, liver, peritoneum, and lungs, which are the favorite sites for GC metastasis owing to the advanced microscopic disease at the time of diagnosis [3]. Bone tends to be the most common metastatic site of breast cancer, prostate cancer, and lung cancer, but GC rarely metastasizes to bone [4]. The dominant incidence of bone metastasis in gastric cancer (GC-BM) patients is only 0.9-10.5% [5,6]; however, this incidence has been found to be as high as 13.4% in autopsies [7,8]. Imaging is the most important method to diagnose GC-BM at present, but elevation of common serum tumor indicators in the blood, such as the carcinoembryonic antigen (CEA), carbohydrate antigen (CA19-9) (CA19-9), and bone-related alkaline phosphatase (ALP), can provide additional diagnostic significance [9,10]. GC-BM usually indicates poor prognosis and short survival [4] and most patients have shorter survival times when experiencing a series of skeletal-related events (SREs) [11].

Bone metastasis is usually associated with the molecules related to bone formation and absorption, such as receptor activator of nuclear factor-κB ligand (RANK-L), parathyroid hormone-related protein (PTHrP), and tumor necrosis factor-alpha (TNF- α), and due to the suitable bone microenvironment, tumor cells can easily settle and proliferate in bone [12]. The metastasis of gastric cancer is mostly related to epithelial-mesenchymal transition (EMT)-related molecular pathways, such as the serine/threonine kinase (AKT) pathway and the Notch pathway [13]. However, there is still no clear evidence that these pathways are associated with GC-BM, and how GC-BM happens at the molecular level is still unknown. GC-BM-specific drugs, therefore, are still to be developed, and platinum drugs and fluorouracil remain the most commonly used first-line chemotherapy regimens, while Trastuzumab is the only targeted drug approved for first-line treatment of HER2-positive metastatic GC [14]. Immune checkpoint inhibitors also demonstrate a potential avenue for the treatment of metastatic GC [15]. Bone-modifying agents (BMAs) such as zoledronic acid and denosumab are beneficial options for alleviating bone symptoms and SREs [16]. This review article discusses the clinical features of GC-BM, as well as its possible molecular pathogenesis pathways and current diagnostic and therapeutic options.

## 2. Bone and Bone Microenvironment

This section reviews bone composition, its transformation, and the bone microenvironment in GC-BM.

### 2.1. Composition of Bone

Bone is a special type of connective tissue that is mainly composed of bone marrow and mineralized extracellular matrix [17]. In the bone marrow, there are two types of stem cells: hematopoietic stem cells and mesenchymal stem cells. Mesenchymal stem cells can differentiate into a variety of non-blood stromal cells, including osteoblasts, osteoclasts, osteocytes, fibroblasts, and adipocytes [18]. Osteoblasts, osteoclasts, and osteocytes are mainly responsible for bone metabolism. Osteoblasts can produce an organic bone matrix composed of type I collagen to mineralize the bone matrix and can be embedded in the matrix in the form of osteocytes to aid in bone formation [19]. The formation of osteoclasts, which are responsible for bone resorption and derived from monocytes in the bone marrow, is closely related to RANK-L, which is expressed on the surface of osteoblasts. Under the stimulation of macrophage-colony stimulating factor (M-CSF), RANK-L promotes the cell fusion of monocytes to form multinucleated osteoclasts [20]. Osteoclasts can bind to the bone matrix through integrins αvβ3, αvβ5, and α2β1 on their membrane surface, and then secrete H+, Cl-, and enzymes, such as cathepsin K and matrix metalloproteinases (MMPs), to degrade bone mineral and extracellular matrix (ECM) bone protein [21]. During degradation, growth factors such as bone morphogenetic proteins (BMPs) and fibroblast growth factors (FGFs) are released from the bone matrix. These factors attract osteoblasts to aggregate and promote new bone formation [22]. Osteocytes regulate the generation and function of osteoblasts and osteoclasts to regulate the dynamic transformation of bone. They secrete the WNT signaling antagonist sclerostin to inhibit osteoblasts and produce M-CSF and monocyte chemoattractant protein-1 (MCP-1) to promote bone resorption [23,24]. The activities of osteoblasts and osteoclasts are constantly tight-coupled to maintain the dynamic balance between bone formation and degradation, but the amount of bone loss cannot be completely replenished, so it is normal for bone mass to decrease with age [25].

### 2.2. Bone Microenvironment and Metastasis

Depending on the functional interaction between tumor cells and osteoclasts or osteoblasts, bone metastasis generally leads to osteolytic destruction or sclerotic hyperplasia [26]. In the area where bone turnover is active, both bone formation and bone resorption can promote the proliferation of tumor cells.

Most osteolytic lesions caused by metastatic tumors are closely related to osteoclasts. Physiologically, osteoclasts are mainly activated by osteoblasts through the RANK-L pathway. However, factors such as IL-6, ICAM-1, Notch ligand, IL-11, PTHrP, and macrophage stimulating protein (MSP) can also activate the RANK-L pathway, resulting in increased osteoclastogenesis [27,28,29,30]. Many tumor cells can also produce PTHrP, IL-1, IL-6, PGE2, TNF, CSF-1, and other factors not only to promote their own survival, but also to increase the production of osteoclasts at the same time. For example, in breast cancer, PTHrP promotes the proliferation of cancer cells as well as osteoclasts production; with the existence of IL-6, which is a potent stimulator of osteoclast formation, the effect of PTHrP on osteoclasts can be significantly enhanced, directly leading to increased osteolysis [31,32]. In the process of osteolysis, growth factors such as transforming growth factor beta (TGF-β), fibroblast growth factors (FGFs), insulin-like growth factors (IGFs), and bone morphogenetic protein-2 (BMP-2) are released into the bone microenvironment to stimulate growth. These factors, in turn, stimulate tumor cells growing, producing, and releasing bone resorption factors [33]. All of them seem to form a closed loop, making the development and destruction of tumors more vicious and unstoppable.

Osteoproliferative lesions are usually caused by abnormal osteoblasts. Osteoblasts can promote tumor cell growth by expressing osteoprotegerin (OPG) and hepatocyte growth factor (HGF) [34,35]. Endothelin-1 (ET-1) released from tumor cells is a key mediator of osteoblastic metastasis; it can inhibit osteoclast bone resorption and osteoclast motility through the endothelin A (ETA) receptor [36]. In prostate cancer, ET-1 also promotes cancer cell proliferation and enhances the mitogenic effects of other growth factors including IGF-1, platelet-derived growth factor (PDGF), and epidermal growth factor (EGF) [37]. Bone morphogenetic proteins (BMPs) belonging to the transforming growth factor beta superfamily can stimulate increased osteoblastogenesis by activating the runt-related transcription factor-2 (RUNX-2) and promote the growth of cancer cells at the same time [38]. Proteolytic enzymes, such as prostate-specific antigens (PSAs) and urokinase-type plasminogen receptors (UPAs), can not only promote the growth of tumor, but also disrupt the balance of osteoblasts and osteoclasts by cleaving PTHrP and promoting TGF-β production, respectively [39,40]. BB isomer, a subtype of platelet-derived growth factor (PDGF), can also stimulate the activation of osteoblasts and promote bone proliferation as a powerful osteogenic factor [33]. This evidence, therefore, illustrates that the progression of osteoblastic lesions can be positively correlated with tumor progression.

Endothelial cells in bone marrow also play an important role in bone metastasis. Tumor cells preferentially attach to bone marrow endothelial cells when entering bone, which is due to the expression of multiple adhesion factors associated with hematopoietic stem cells (HSCs) including selectin and Galectin-3 [41]. Tumor cells can also express receptors for ligands expressed by endothelial cells, such as CD44 and integrins, to enhance their adhesion [42,43]. In addition, tumor cells can increase angiogenesis by inducing the increase of vascular endothelial growth factor (VEGF) which is downstream of hypoxia-inducible factor 1α (HIF-1α) and PTHrP expression to form a blood-rich environment conducive to tumor growth [44].

In the bone marrow, cytotoxic T cells and natural killer (NK) cells are under-expressed relative to immunosuppressive regulatory T cells (Tregs) and myeloid-derived suppressor cells (MDSCs) [45]. B cells and T cells can produce cytokines such as OPG and RANK-L to promote bone resorption and inhibit osteoclast activity under different conditions [46]. Tumor necrosis factor alpha (TNF-α), produced by T cells, can stimulate osteoclastogenesis, while interferon gamma (IFN-γ) produced by T cells can have the opposite effect by inhibiting RANK-L [47,48]. T cell function can also be affected by metastatic cancer cells. The enhanced bone resorption caused by metastatic tumor cells leads to a large amount of transforming growth factor-β (TGF-β) released from bone, resulting in the inhibition of T cell and NK cell proliferation and function [49]. Helper T cells 2 (Th2) can promote bone formation through the production of PTHrP, whereas Helper T cells 1 (Th1) can be inhibited by IFN-γ [50]. MDSCs can directly act as osteoclast precursors to promote osteolytic lesions [51], while Treg cells are associated with osteoclast inhibition [52]. Tumor-associated macrophages (TAMs) can secrete IL-2, IFN, and IL-12 to kill tumor cells, but they can also secrete a variety of angiogenic factors and cytokines to promote the growth of tumor cells and IL-10 to inhibit the antitumor response of cytotoxic T cells [53]. Overall, the bone marrow microenvironment provides a very suitable place for tumor metastasis. Even though anti-tumor immunity may play a certain role in the early stage, the formation of immune escape and further tumor progression can rarely be avoided in the end [54] (Figure 1).

Exosomes are widely considered to have an important influence on gastric cancer growth and metastasis [55]. They can secrete various ncRNAs, which mainly are miRNAs such as miR-217, miR-423-5p, and miR-191 to regulate the expression of tumor-related genes and molecules to promote EMT [56,57,58]. Additionally, tumor exosomes (TEXs) can present TGF-β directly and start TGF-β/Smad cascade reactions to transform umbilical cord MSCs into the CAFs. They can also secrete miR-27a to transform residential fibroblasts into CAFs and then promote tumor progression [59]. Tumor exosomes (TEXs) can also affect the functions of immune cells. They can induce macrophages to differentiate into M2-type, in which they produce exosomes containing apolipoprotein E to activate PI3K-AKT signaling pathways and then promote metastasis [60]. Exosomes secreted by GC cells can guide monocytes into M2-characterized programmed cell death 1 (PD-1)-positive tumor-associated macrophages, upregulating the secretion of IL-10 and thus affecting the function of CD8+T cells [61]. Exosomes secreted by bone marrow-derived MSCs are also closely related to tumor development. Studies have found that they can regulate tumor proliferation and progression by activating the Hedgehog signaling pathway through miR-221 [62]. Although increasing research has proved the tumor-promoting effect of exosomes, specific types of exosomes that participate in the GC-BM process remain to be solved.

Signaling pathways such as the PI3K/AKT pathway, Notch pathway, TGF-β pathway, and NF-κB signaling pathway are the main factors that closely relate to gastric cancer metastasis [63,64,65]. At present, the molecular mediators related to bone metastasis of solid tumors are mainly RANK/RANK-L/OPG, TNF- α, and PTHrP [66,67]. RANK-L, which belongs to the tumor necrosis factor (TNF) family, can be secreted not only by bone-related cells but also by infiltrating T cells, which are abundant in gastric cancer [68]. Wang et al. found that c-Src can mediate the activation and interaction of Cav-1 with RANK to induce RANK-L to activate the PI3K/AKT and ERK pathways to promote gastric cancer cell migration [69]. However, D’Amico et al. found that the bone metastasis of gastric cancer does not depend on the RANK-L mechanism [70]. At present, the question of GC-BM pathogenesis at the molecular mechanism level remains unsolved and the “true face” of the disease has eluded science, thus compromising the development of diagnostic and therapeutic options.

## 3. Clinical Bone Metastasis of Gastric Cancer

### 3.1. Clinical Features

Gastric cancer rarely metastases to bone, and when GC-BM happens to patients, they may have no obvious symptoms, or bone pain is the only symptom. Retrospective analyses of different gastric cancer populations and the statistics of case studies show that the incidence of bone metastasis in GC patients is roughly in the range of 0.9–13.4% [5,6,8,10,71]. GC-BM can develop in cancers regardless of the tumor stage; however, it is more prevalent in advanced-stage disease. The proportion of patients with stage IV gastric cancer with bone metastasis exceeds the sum of stage I-III patients [72], and the risk of bone metastasis in gastric cancer patients with lung or liver metastasis is significantly increased [73]. Park et al. found that even after radical resection of gastric cancer, 1.8% of patients still had a recurrence of bone metastasis, and the median time from surgery to the discovery of recurrent bone metastases was only 28 months [72].

The incidence of gastric cancer bone metastases did not differ significantly among ethnic groups, but in terms of gender, the results of different population studies were not identical. According to the statistics of Qiu et al., there is no significant difference in the incidence of bone metastasis between men and women [73]. Yoshinori’s retrospective study also found that the proportion of men and women was not significantly different [10]. However, the study of Silvestris et al. found that men are about 32% more at risk than women of developing GC-BM [74]. In terms of the age of onset, most studies show that the median age is less than 65 years old, and the median survival time of overall survival (OS) after the diagnosis of bone metastasis is as short as about 3 months and as long as 14 months, which is much shorter than that of patients with liver and lung metastasis [10,73,74,75,76,77,78,79].

In GC-BM, the primary site is more likely to be in the cardiac and gastric body than that in the antrum and pylorus [80]. The most common site for GC-BM is the spine, followed by pelvis, ribs, sternum, and long bones of limbs [10,73,78]. Patients with bone metastases also have a higher risk of brain and lung metastases [80]. Poorly differentiated tumors tend to have the highest rate of GC-BM versus well-differentiated tumors, while moderately differentiated tumors are more likely to metastasize to the liver, lung, and brain [73]. The most common pathological types of primary lesions in GC-BM are adenocarcinoma and poorly-differentiated carcinoma [5]. The incidence rate of signet ring cell carcinoma is low, but due to its high level of malignancy, the bone metastasis rate of signet ring cell carcinoma is much higher than that of adenocarcinoma [5]. According to Lauren’s classification, diffuse type GC is the most common type to progress to GC-BM while intestinal type GC is relatively rare [76]. About 80% of patients with bone metastasis experience multiple bone metastases, most of which are distributed axially. Most bone metastases are metachronous tumors, and although synchronous tumors are relatively rare, the OS of patients with them are shorter than that of patients with metachronous tumors [76]. In addition, more than half of the lesions caused by GC-BM are osteolytic lesions, followed by mixed lesions, and osteogenic lesions are the least common [10,81].

Skeletal-related events (SREs) caused by bone metastasis include pathological fractures, hypercalcemia, spinal cord injury, uncontrollable pain requiring bone surgery or radiotherapy, and bone injury caused by surgery or radiotherapy and chemotherapy [82]. Among them, radiotherapy bone injury is the most common SRE, followed by pathological fracture, bone surgery bone injury, spinal cord compression, and hypercalcemia [74]. Once bone metastases occur, osteolytic or osteogenic lesions can easily cause SREs and ~31% of gastric cancer patients with bone metastases experience SREs [74]. The poor prognosis of GC-BM patients is further comprised by SREs, and the median survival time of patients with SREs is significantly shorter than non-SRE patients [11]. When metastases invade bone, they also often causes hematological abnormalities, and in severe cases, they even cause the occurrence of disseminated intravascular coagulation (DIC), resulting in rapid deterioration of the disease and death within 2 months [10]. Although the specificity of clinical characteristics of GC-BM is not precise, the impact on the prognosis and quality of life of patients should be given significant consideration.

### 3.2. Diagnosis Methods

Pathological examination is the gold standard for the diagnosis of almost all tumors, including bone tumors. However, imaging methods and examination of tumor indicators are also of great significance for the diagnosis of tumors. The primary diagnosis of bone metastasis is mostly based on bone-related clinical manifestations such as bone pain and neuromuscular symptoms, and to date, bone radionuclide scans are the most sensitive examination method to diagnose bone tumors. However, for GC, the European Society of Medical Oncology (ESM) and National Comprehensive Cancer Network (NCCN) guidelines do not recommend routine bone radionuclide scans in patients during diagnosis or treatment [83,84]. Computed Tomography (CT) or enhanced CT is mainly used for the diagnosis and monitoring of primary lesions of gastric cancer and metastasis of other organs. Currently, it is also not a routine examination method for bone metastasis and is only recommended when there are symptoms [5]. This may lead to the asymptomatic bone metastasis of GC being largely neglected. Magnetic resonance imaging (MRI) is used only in the presence of neurological symptoms and in the assessment of spinal and pelvic metastasis [85]. 18fluorodeoxiglucose Positron Emission Tomography–Computed Tomography (18FDG-PET-CT) has a certain diagnostic significance for evaluating the systemic conditions of most patients with bone metastasis of GC, including primary lesions and metastatic lesions. However, due to the low expression of glucose transporter 1 (GLUT1) in the diffuse tissue type of gastric cancer cells, 18FDG-PET-CT is less sensitive to this type of patient [86].

A special phenomenon called bone flare may occur during the treatment of bone metastasis. Bone flare refers to the false image of “progression of bone metastases” caused by the increased uptake of radiotracer on the bone scan when the primary tumor and bone metastases are treated effectively [87]. Bone flare frequently occurs in osteoblastic lesions, which can make it difficult to identify progressive bone metastases. Although this phenomenon is rare, it has occurred in GC [88]. Therefore, such phenomena should be carefully considered in clinical evaluation of the disease for effective patient care.

The continuous increase in the levels of tumor markers, such as CEA, CA19-9, and CA72-4, generally indicates the occurrence and progression of tumors. These biomarkers often increase in bone metastasis as well; however, their specificity is limited [10]. Bone ALP is the most significant biological marker in bone metastasis of GC [9]. Bone ALP is a specific marker of osteoblast metabolism, which is significantly correlated with the presence of bone metastasis and the degree of bone involvement in metastatic tumors, and bone ALP has been proved to be a significant predictor of bone metastases in breast and prostate cancer patients [89]. More than half of patients with GC-BM show elevated ALP and tumor markers [9,10], suggesting that GC patients presenting with elevated tumor markers and ALP should be evaluated and examined for GC-BM even if they have no obvious symptoms. (Table 1)

### 3.3. Treatment

The treatment of bone metastasis of gastric cancer mainly includes two aspects: the treatment of tumors and the treatment of bone-related SREs. For metastatic gastric cancer, palliative care and individualized treatment are mostly given, aiming to ultimately control symptoms, control progression, and prolong life. Chemotherapy, targeted therapy, and immunotherapy are the current mainstream non-surgical treatment methods [2]. At present, there is no universal standard treatment method in the world. For example, there are differences between Japan and other countries on whether 5-FU is used in combination [101,102]. Oral fluorouracil S-1 is one of the most widely studied chemotherapy drugs for gastric cancer in Japan. The early phase II study of S-1 monotherapy in metastatic gastric cancer shows that the effective response rate can be as high as 49% [103]. It was also found that S-1 combined with platinum drugs can significantly prolong the median overall survival in the phase III study [104]. The United States usually uses CF regimen (cisplatin plus 5-FU), ECF regimen (epirubicin plus cisplatin and 5-FU), and DCF regimen (docetaxel, cisplatin, and 5-FU) to treat progressive and metastatic gastric cancer [105] (Table 2).

According to the characteristics of patients in China, the current first-line treatment regimens vary according to whether HER2 is expressed or not. When HER2 is positive, the first choice is trastuzumab combined with oxaliplatin/cisplatin and 5-FU/capecitabine/tegafur. On the contrary, when HER2 is negative, oxaliplatin/cisplatin/paclitaxel/docetaxel combined with fluorouracil (5-FU/capecitabine/tegafur) becomes the first choice. The second choice is DCF or modified DCF regimens, which are mostly suitable for patients with fair physical conditions and a high tumor load. Second-line treatment is mostly monotherapy and commonly used drugs, mostly including those not used in first-line treatment. Trastuzumab combined with other first-line chemotherapy drugs is recommended for patients who have not used trastuzumab before. For patients with positive MSI-H and PD-1/PD-L1 expression, the combination of anti-PD-1/PD-L1 monoclonal antibodies such as pembrolizumab, nivolumab, and chemotherapy drugs is recommended. Anti-angiogenic targeted drugs such as apatinib and bevacizumab are also recommended as the first choice for third-line therapy [107].

Nowadays, there are also studies experimenting on the efficacy of drug combinations that are not mentioned in the guidelines. Various studies have compared the efficacy of the FLO (fluorouracil, leucovorin, oxaliplatin) regimen and the FLP (fluorouracil, leucovorin, cisplatin) regimen as first-line treatments for metastatic gastric cancer, but specific data and evaluations on the efficacy of bone metastasis are needed [108]. In addition, ramucirumab, an anti-VEGFR-2 monoclonal antibody that has been proved to improve survival as a monotherapy, has been used in combination with paclitaxel for the second-line treatment of metastatic gastric cancer [109], but they have not been specifically tested for their efficacy in GC-BM. Furthermore, some studies have tested the effects of different courses of cisplatin +s-1 on the efficacy of metastatic gastric cancer and bone metastasis [110], but detailed prognostic data remains elusive. Only a single study indicates that the CX regimen (cisplatin combined with capecitabine) in GC-BM has a different outcome on bone metastases compared with the CF regimen (cisplatin and fluorouracil) [111]. (Table 3)

Treatments of symptoms caused by bone metastases and SREs are also important. Patients with bone metastasis often have unbearable bone pain, and opioids or other painkillers can alleviate the pain of patients. If painkillers fail, radiotherapy can effectively control the pain, especially neuropathic pain [124]. For some patients, necessary surgery can be of great help in controlling pain and avoiding fractures and spinal cord compression [11]. Bisphosphonates such as zoledronic acid can inhibit osteoclast function and induce osteoclast apoptosis [125]; it can also regulate the activity of matrix metalloproteinases, change the bone matrix environment, and induce tumor cell apoptosis to inhibit bone metastasis [126]. Denosumab, a bone resorption inhibitor, is a monoclonal antibody against RANK-L; it can block the binding of RANK and RANK-L, thereby inhibiting the differentiation of osteoclasts [127]. It was found that denosumab was superior to zoledronic acid in delaying the time of the first SRE occurrence and reducing the damage caused by SREs [16]. Hypercalcemia caused by osteolytic lesions requires a lot of fluid replacement and diuresis, but with bisphosphonate treatment, it can be more effectively relieved [16]. Osteoblastic lesions can cause excessive calcium deposition and severe hypocalcemia, which is called “hungry bone” syndrome. When this syndrome occurs, timely calcium supplementation is needed to correct hypocalcemia and attention needs to be paid to whether there is hypomagnesemia [128]. Despite these developments, the existing treatment options for GC-BM remain limited and there is a need to explore other clinical and molecular mechanisms to develop a more effective targeted treatment.

## 4. Conclusions

As one of the deadly malignant tumors, GC can metastasize to any sites at any stage. Although the incidence of GC-BM is not as high as that of cancers of the peritoneum, liver, and lung, the bone microenvironment provides an option for gastric tumor cells’ colonization and proliferation. As an important pathway in bone homeostasis, the RANK/RANK-L/OPG pathway can be abnormally activated by cytokines secreted from tumor cells such as PTHrP, IL-6, and TNF that can promote osteolytic changes. During the process of osteolysis, growth factors such as TGFs and FGFs are released from the bone, and in turn, these factors promote tumor growth and cytokine expression. All of these constitute a perfect loop that makes the tumor progression and osteolysis overwhelming. Other factors such as ET-1, BMPs, PSA, UPA, and PDGF not only play important roles in excessive osteogenesis, but also in tumor cell proliferation and development. The interaction among osteoblasts, osteoclasts, and tumor cells finally leads to bone abnormalities and tumor development. Although immune cells in the bone microenvironment can initially suppress pathogenesis to some degree, the increased expression of TGF-β and TAMs and more immunosuppressive cells induce increasing levels of immunotolerance. A further complication is that the molecular pathogenesis pathways of GC-BM are still unclear, despite being identified in other cancers. For example, despite increased expression of the RANK-L, PTHrP, BMP4, and AKT pathways in GC, there is no conclusive evidence to show that these molecules’ expressions directly influence the development of GC-BM.

Clinical evidence indicates GC-BM is most common in patients with stage IV, poorly-differentiated GC, gastric adenocarcinoma, and diffuse gastric cancer. The median OS of GC patients with bone metastasis is about 8 months, which is much shorter than that of GC patients with liver and lung metastasis. Most commonly, the spine is the site chosen for BM, as well as the brain and lungs, and the OS of GC-BM patients is most likely compromised by SREs caused by bone lesions and inactive inspection.

The treatment options and curative effects for GC-BM remain limited because its molecular pathways are not fully understood. Although traditional chemotherapy and developing targeted drugs and immunotherapy can alleviate GC-BM symptoms in some patients, further research is required to develop more effective and precise drugs with fewer side effects.

## Figures and Tables

**Figure 1 cancers-14-04888-f001:**
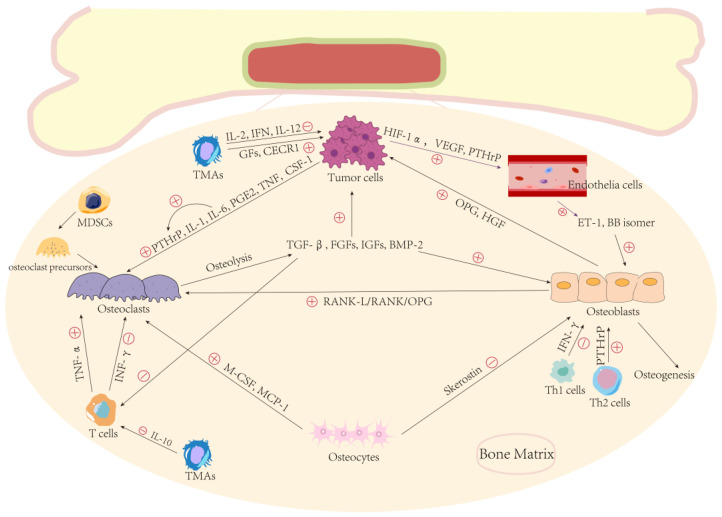
In the bone matrix, osteoclasts, osteoblasts, and osteocytes restrict and balance each other by multiple cytokines such as RANK-L, M-CSF, and skerostin. Once the tumor cells invade, the balance is disrupted by various factors secreted by tumor cells such as PTHrP, IL-6, HIF-1α, and CSF-1. With the broken balance, osteolytic or osteoblastic lesions increase abnormally and abundant factors such as GFs, TGF-β, and BMPs, which can promote tumor cell proliferation, are released from the bone matrix. T cells in the bone matrix can both suppress and promote osteoclasts under different conditions through TNF-α and INF-γ, respectively. The function of T cells can be repressed by increased TGF-β and IL-10 released from TMAs, which can promote and kill tumor cells through secreting different factors. MDSCs can act as osteoclast precursors to increase osteolysis. Additionally, Th1 cells repress the function of osteoblasts through IFN-γ while Th2 cells promote it through PTHrP.

**Table 1 cancers-14-04888-t001:** Possible increased biomarkers in serum test of GC-BM.

Markers	Signification of Increase	Application
Bone alkaline phosphatase (BALP) a [90,91,92,93]	A bone formation marker. High levels of it are associated with increased risks for all negative clinical outcomes, including a shorter time of a first SRE, disease progression, and death. Compared with high NTX, less increased risk associated with high bone-specific alkaline phosphatase levels.	Diagnosis and prognosis of BM from solid tumors. Prognosis skeletal-related events. Prognosis during antiresorptive therapy. Prediction of response to treatment.
N-telopeptide (NTX) b [92,93,94,95]	A bone resorption marker. High baseline levels are associated with a significantly increased risk of SREs, bone disease progression, and death.	Prognosis of BM from solid tumors. Prognosis skeletal-related events. Prognosis during antiresorptive therapy. Prediction of response to treatment.
Receptor activator of nuclear factor κB-ligand/osteoprotegerin (RANKL/OPG) [96,97,98]	Bone resorption markers. In severe osteolysis, RANKL Expression and RANKL/OPG mRNA Ratio are significantly increased, and the severity of osteolysis is correlated with the increase of serum RANKL and RANKL/OPG levels.	Diagnosis of bone metastasis in solid tumors

a. BALP is positively associated with ALP, osteocalcin, CA19-9, and CA72-4 [9]. The level of serum BALP in patients with BM was significantly higher than that in patients without bone lesions [99]. The high level of bone BALP is also significantly related to the occurrence of SRE [92]. Survival is improved with atrasentan compared to placebo in patients with top 25% high BALP [91]. b. In patients not receiving bisphosphonate therapy, NTX relates to increased risks of negative clinical outcomes such as SREs, disease progression, and death [93]. High level of NTX is associated with a twofold increased risk of skeletal complications [92]. Compared with the survival of patients with persistent elevated NTX levels, that of patients with normalized NTX level during zoledronic acid treatment is improved [100].

**Table 2 cancers-14-04888-t002:** Recommended treatments of metastasis gastric cancer in United States and Japan.

	United States [83]	Japan [106]
First Line	HER2(+): Trastuzumab+ Fluoropyrimidine+ Cisplatin/Oxaliplatin HER2(-): ① Nivolumab + Fluoropyrimidine +Oxaliplatin (PD-L1 CPS of ≥5) ② Fluoropyrimidine + Oxaliplatin/Cisplatin. ③ Capecitabine + Oxaliplatin ④ Irinotecan + Fluorouracil ⑤ mDCF regimens (Docetaxel + Oxaliplatin + Calcium Folinate + Tegafur)	HER2(+): trastuzumab+ cisplatin+ capecitabine/S-1 HER2(-): ① S-1+ Cisplatin/Oxaliplatin ② Capecitabine + Cisplatin/Oxaliplatin ③ 5-FU + Levofolinate calcium + Oxaliplatin
Second Line	① Ramucirumab + Paclitaxel ② Fam-trastuzumab deruxtecan-nxki (for patients with HER2+ and had received prior trastuzumab-based therapy) ③ Monotherapy: Docetaxel, Paclitaxel, and Irinotecan ④ Irinotecan+Fluorouracil/Cisplatin/Ramucirumab/Docetaxel ⑤ Pembrolizumab (MSI-H/dMMR Tumors) ⑥ Ntrectinib/Larotrectinib (NTRK gene fusion-positive tumors)	Ramucirumab + Paclitaxel
Third Line	-	Nivolumab/Irinotecan

**Table 3 cancers-14-04888-t003:** Recommended treatment for metastatic gastric cancer in China [107].

	First Line a	Second Line b	Third and Above Linesc
First choice	HER2(+):①Trastuzumab + Oxaliplatin + 5-FU/Capecitabine ②Trastuzumab +Cisplatin+ 5-FU/Capecitabine HER2(-):①Fluorouracil (5-FU/capecitabine/tegafur)+oxaliplatin/cisplatin ②Fluorouracil (5-FU/capecitabine/tegafur)+paclitaxel/docetaxel	Monotherapy: Paclitaxel/Docetaxel/Irinotecan	Anti-angiogenic targeted drugs: Apatinib, Bevacizumab
Second choice	HER2(+): Trastuzumab + Fluorouracil (5-FU/capecitabine/tegafur)+ Oxaliplatin/Cisplatin HER2(-): ①DCF regimens (Docetaxel + cisplatin + 5-FU) ②mDCF regimens (Docetaxel + Oxaliplatin + Calcium Folinate + tegafur)	①Trastuzumab + paclitaxel ②Paclitaxel/Docetaxel + Fluorouracil ③Pembrolizumab (for patients with MSI-H, PD-1/PD-L1 positive)	Pembrolizumab (for patients with MSI-H, PD-1/PD-L1 CPS ≥ 1)
Third choice	①Trastuzumab + first-line chemotherapy regimens ②pembrolizumab monotherapy (PD-L1 CPS≥1)	Trastuzumab + other lines chemotherapy regimens	Single drug chemotherapy (refer to the second-line recommendations)

a. In the treatment of HER2+ metastatic gastric cancer, except trastuzumab, other HER2-targeted drugs including pertuzumab and lapatinib have no positive response [112,113]. ToGA test found that trastuzumab combined with first-line chemotherapy could improve OS and PFS of HER2+ first-treated advanced metastatic gastric cancer patients and showed good tolerance, effectiveness, and safety compared with chemotherapy alone [114,115]. For HER2– metastatic gastric cancer, based on Chinese data research, it is recommended to use fluoropyrimidine combined with platinum dual drug therapy, especially combined with oxaliplatin, as the probability of adverse events is significantly lower than that when combined with cisplatin [116]. Paclitaxel combined with fluorouracil has also shown sufficient efficacy and safety in clinical research and practice [117]. The clinical application of DCF regimens is limited due to its high toxicity, while the improved mDCF regimen is relatively more effective and more tolerant [118,119]. For patients with PD-L1 CPS ≥ 1, the study shows that the OS of patients after single-drug therapy with pembrolizumab is not worse than that of chemotherapy, but single drug immunotherapy is not recommended because of the lack of sufficient data on the risks [15]. b. Single drug treatment is recommended by second-line chemotherapy; however, for the elderly and infirm patients, studies showed that dual drugs with an appropriate reduction of chemotherapy dosage are harmless to PFS and have a better overall treatment effect [120]. Clinical trials of dMMR/MSI-H malignant tumors including gastric cancer that failed conventional treatment showed that pembrolizumab may be beneficial and safer than chemotherapy [121]. c. Apatinib mesylate (VEGFR2 small molecule tyrosine kinase inhibitor), which is a commonly used anti-angiogenic drug for patients with advanced gastric cancer, can extend mPFS and has been approved for the third or higher lines treatment of patients with advanced gastric cancer or gastroesophageal junction (EGJ) adenocarcinoma [122]. Studies have shown that pembrolizumab can also be used as a third-line therapy for recurrent or metastatic adenocarcinoma of gastric cancer and EGJ cancer with PD-L1 CPS ≥ 1 to prolong the survival temporarily [123].

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
