# Peer review of "Tumor Microenvironment, Clinical Features, and Advances in Therapy for Bone Metastasis in Gastric Cancer"

_cancers, 2022, doi:10.3390/cancers14194888_

Round 1
Reviewer 1 Report
Evaluation of ” Tumor microenvironment, clinical features, and advances in therapy for
bone metastasis in gastric cancer” by Sun et al. to Cancers
In this review manuscript authors present various factors contributing to the bone metastasis formation in gastric cancer as well as metastatic gastric cancer treatment options.
-The manuscript contains a lot of data, but since it is not very well organized, it is rather hard to get much out of it. In addition to that, the language at some parts is of rather poor quality, which makes it difficult to understand what authors want to present. For example, the subtitle at lane 81: what authors mean with “dynamic transformation of bone”. The actual following chapter tells nothing about “dynamic transformation”. And so on.
-The figure 1 is faint and the text inside the figure is in many cases too small to read. It might be more useful if this figure would compare normal bone matrix to bone matrix with cancer invasion.
-There are some strange references in it such as “LUCIA et al” at the lane 193 and “silvestris et al” at the lane 216. Some parts of the text seem as it has been copy pasted from another text that has not been written carefully.
-The text would require serious reconsideration from authors of what they really want to say. For example, in the lane 231 “The spine is the most easily metastatic site”. Lane 254 “…clinical manifestations, imaging and blood tests…
-Since this is a feature review on Cancer Therapy, the table 1 “Tretment for metastatic gastric cancer in China” should rather be about comparison of metastatic gastric cancer treatments in for example USA, China and Japan, since authors seem to have information on them since they mention them in the body text. The table 1 should anyway be reorganized, since it is impossible to understand what it presents. Perhaps it should be made into a figure rather than table.
-Under diagnosis methods authors should make a table of bone metastasis markers and their usability.
Author Response
Dear Editors and Reviewers,
Thank you for your letter and the reviewers’ comments concerning our manuscript (cancers-1933706) entitled “Tumor microenvironment, clinical features, and advances in therapy for bone metastasis in gastric cancer”. We believe your valuable comments have improved the quality of the paper considerably. We have read through comments carefully and have made the requested changes. Based on the instructions provided in your letter, we have uploaded the file of the revised manuscript. Revisions in the text are highlighted in red font for easy readability.
We thank you for allowing us to resubmit a revised copy of the manuscript and we highly appreciate your time and consideration. Should you have any remaining questions, please do not hesitate to contact me.
Best regards,
Sincerely yours,
Dr. Xiaodong Li
Reviewer 1:
Evaluation of “Tumor microenvironment, clinical features, and advances in therapy for bone metastasis in gastric cancer” by Sun et al. to Cancers
In this review manuscript authors present various factors contributing to the bone metastasis formation in gastric cancer as well as metastatic gastric cancer treatment options.
- The manuscript contains a lot of data, but since it is not very well organized, it is rather hard to get much out of it. In addition to that, the language at some parts is of rather poor quality, which makes it difficult to understand what authors want to present. For example, the subtitle at lane 81: what authors mean with “dynamic transformation of bone”. The actual following chapter tells nothing about “dynamic transformation”. And so on.
Reply: Thank you for your valuable comments. We have reorganized the text and polished the language. In response to your query regarding dynamic transformation for this sub-title we have attempted to specifically address this in GC-BM, for example in this section the text illustrates “Osteoclasts can bind to bone matrix through integrins αvβ3, αvβ5, and α2β1 on their membrane surface, and then secrete H+, Cl-, and enzymes, such as cathepsin K and matrix metalloproteinases (MMPs), to degrade bone mineral and extracellular matrix (ECM) bone protein. During degradation, growth factors such as bone morphogenetic proteins (BMPs) and fibroblast growth factors (FGFs) are released from bone matrix. Besides, subtitles have been revised to make them correspond to the subsequent text in paragraph.
- The figure 1 is faint and the text inside the figure is in many cases too small to read. It might be more useful if this figure would compare normal bone matrix to bone matrix with cancer invasion.
Reply: The quality of figure has been improved, and font size inside the figures has been increased. The normal bone matrix has been described.
- There are some strange references in it such as “LUCIA et al” at the lane 193 and “silvestris et al” at the lane 216. Some parts of the text seem as it has been copy pasted from another text that has not been written carefully.
Reply: The whole text and references have been revised.
- The text would require serious reconsideration from authors of what they really want to say. For example, in the lane 231 “The spine is the most easily metastatic site”. Lane 254 “…clinical manifestations, imaging and blood tests…” in line 273-5.
Reply: The description of the sentences has been revised to make the text more clarified.
- Since this is a feature review on Cancer Therapy, the table 1 “Treatment for metastatic gastric cancer in China” should rather be about comparison of metastatic gastric cancer treatments in for example USA, China, and Japan, since authors seem to have information on them since they mention them in the body text. The table 1 should anyway be reorganized, since it is impossible to understand what it presents. Perhaps it should be made into a figure rather than table.
Reply: Table on the treatments in the USA and Japan have been added(Please see Table 2). The tables have been reorganized.
- Under diagnosis methods authors should make a table of bone metastasis markers and their usability.
Reply: A table of markers has been added. Please see Table 1.
Reviewer 2 Report
The authors have performed a review article about the role of tumor microenvironment in gastric cancer progression, bone metastasis and response to therapy. They discuss also therapeutic strategies to improve the management of bone metastasis in patients with gastric cancer.
The paper is clear, interesting for readers and written in a good English.
Nevertheless, I have one constructive comment: the role of exosomes and microRNAs should be discussed.
Author Response
Dear Editors and Reviewers,
Thank you for your letter and the reviewers’ comments concerning our manuscript (cancers-1933706) entitled “Tumor microenvironment, clinical features, and advances in therapy for bone metastasis in gastric cancer”. We believe your valuable comments have improved the quality of the paper considerably. We have read through comments carefully and have made the requested changes. Based on the instructions provided in your letter, we have uploaded the file of the revised manuscript. Revisions in the text are highlighted in red font for easy readability.
We thank you for allowing us to resubmit a revised copy of the manuscript and we highly appreciate your time and consideration. Should you have any remaining questions, please do not hesitate to contact me.
Best regards,
Sincerely yours,
Dr. Xiaodong Li
Reviewer 2
The authors have performed a review article about the role of tumor microenvironment in gastric cancer progression, bone metastasis and response to therapy. They discuss also therapeutic strategies to improve the management of bone metastasis in patients with gastric cancer.
The paper is clear, interesting for readers and written in a good English.
Nevertheless, I have one constructive comment: the role of exosomes and microRNAs should be discussed.
Reply: Thank you for your valuable comments. We have now included a discussion on the role of exosomes and miRNAs. Please see page 5, lines 185-201.
Reviewer 3 Report
This review article summarizes the clinical features of gastric cancer bone metastasis (GC-BM) and its possible molecular pathogenesis pathways and current diagnostic and therapeutic options. Gastric cancer is predominant cancer affecting close to a million lives, this article sheds light on a subtype of GC, which is significant to the field. Although the incidence of GC-BM in particular, is not as high (0.9% - 13.4%) as that in cancers of the peritoneum, liver and lung, the authors pinpoint that the bone microenvironment provides an option for gastric tumor cells’ colonization and proliferation.
After discussing the epidemiology of GC-BM briefly, the authors pinpoint the treatment options and disease-modifying agents available to achieve clinical benefits. This manuscript reviews the bone composition, its transformation, and the bone microenvironment in GC-BM. This review explains the clinical bone metastasis of gastric cancer, skeletal-related events (SREs) associated with GC-BM, diagnosis methods and treatment methods in a coherent order.
This work highlight the lack of clear evidence to associate GC-BM and BM-associated pathways (could be RANK-L, TNF etc) and claim the need to understand the molecular working mechanism of GC-BM.
Some minor changes are needed which include modifying the figures (pixelated, use better illustrator) There is a reference missing on Page 4, line no 173 reference, kindly add that.
The authors have shown a table with the treatment options for metastatic gastric cancer in China (briefly mentioned treatment options in Japan and USA). This table requires citations for their claims and please differentiate what they mean by first choice or first-line treatment. Kindly DO NOT remove this table, it's highly informative and useful.
The language of communication is highly scientific and comprehensible. This review article captures the key publications in the GC field and is of good quality for publication. I recommend this work for publication with minor changes. Thank you.
Author Response
Dear Editors and Reviewers,
Thank you for your letter and the reviewers’ comments concerning our manuscript (cancers-1933706) entitled “Tumor microenvironment, clinical features, and advances in therapy for bone metastasis in gastric cancer”. We believe your valuable comments have improved the quality of the paper considerably. We have read through comments carefully and have made the requested changes. Based on the instructions provided in your letter, we have uploaded the file of the revised manuscript. Revisions in the text are highlighted in red font for easy readability.
We thank you for allowing us to resubmit a revised copy of the manuscript and we highly appreciate your time and consideration. Should you have any remaining questions, please do not hesitate to contact me.
Best regards,
Sincerely yours,
Dr. Xiaodong Li
Reviewer 3
This review article summarizes the clinical features of gastric cancer bone metastasis (GC-BM) and its possible molecular pathogenesis pathways and current diagnostic and therapeutic options. Gastric cancer is predominant cancer affecting close to a million lives, this article sheds light on a subtype of GC, which is significant to the field. Although the incidence of GC-BM in particular, is not as high (0.9% - 13.4%) as that in cancers of the peritoneum, liver and lung, the authors pinpoint that the bone microenvironment provides an option for gastric tumor cells’ colonization and proliferation.
After discussing the epidemiology of GC-BM briefly, the authors pinpoint the treatment options and disease-modifying agents available to achieve clinical benefits. This manuscript reviews the bone composition, its transformation, and the bone microenvironment in GC-BM. This review explains the clinical bone metastasis of gastric cancer, skeletal-related events (SREs) associated with GC-BM, diagnosis methods and treatment methods in a coherent order.
This work highlights the lack of clear evidence to associate GC-BM and BM-associated pathways (could be RANK-L, TNF etc) and claim the need to understand the molecular working mechanism of GC-BM.
Some minor changes are needed which include modifying the figures (pixelated, use better illustrator) There is a reference missing on Page 4, line no 173 reference, kindly add that.
The authors have shown a table with the treatment options for metastatic gastric cancer in China (briefly mentioned treatment options in Japan and USA). This table requires citations for their claims and please differentiate what they mean by first choice or first-line treatment. Kindly DO NOT remove this table, it's highly informative and useful.
The language of communication is highly scientific and comprehensible. This review article captures the key publications in the GC field and is of good quality for publication. I recommend this work for publication with minor changes. Thank you.
Reply: Thank you for your kind comments. The quality of figure has been improved (please see figure 1). The reference has been added to page 4, line no. 171. Further, we have revised the sentence (please page 4, line 171) and more details have been added to Table in response to the Reviewers comments.
Round 2
Reviewer 1 Report
The figure 1 is still of poor quality. Use adobe illustrator or similar to re-make it. Please carefully double-check that the information give in tables has no mistakes.
